# DECOUPLE-THEN-MERGE:
# TOWARDS BETTER TRAINING FOR DIFFUSION MODELS

## ABSTRACT

Diffusion models are trained by learning a sequence of models that reverse each step of noise corruption. Typically, the model parameters are fully shared across multiple timesteps to enhance training efficiency. However, since the denoising tasks differ at each timestep, the gradients computed at different timesteps may conflict, potentially degrading the overall performance of image generation. To solve this issue, this work proposes a **De**couple-then-**Me**rge (**DeMe**) framework, which begins with a pretrained model and finetunes separate models tailored to specific timesteps. We introduce several improved techniques during the finetuning stage to promote effective knowledge sharing while minimizing training interference across timesteps. Finally, after finetuning, these separate models can be merged into a single model in the parameter space, ensuring efficient and practical inference. Experimental results show significant generation quality improvements upon 6 benchmarks including Stable Diffusion on COCO30K, ImageNet1K, PartiPrompts, and DDPM on LSUN Church, LSUN Bedroom, and CIFAR10. Code is included in the supplementary material and will be released on Github.

## 1 INTRODUCTION

Generative modeling has seen significant progress in recent years, primarily driven by the development of Diffusion Probabilistic Models (DPMs) (Ho et al., 2020; Nichol & Dhariwal, 2021; Rombach et al., 2022b). These models have been applied to various tasks such as text-to-image generation (Rombach et al., 2022a), image-to-image translation (Saharia et al., 2022a), image editing (Yang et al., 2023a), and video generation (Ho et al., 2022; Blattmann et al., 2023), yielding excellent performance. Compared with other generative models such as variational auto-encoders (VAEs) (Kingma & Welling, 2013), and generative adversarial networks (GANs) (Goodfellow et al., 2014), the most distinct characteristic of DPMs is that DPMs need to learn a *sequence* of models for denoising at multiple timesteps. Training the neural network to fit this step-wise denoising conditional distribution facilitates tractable, stable training and high-fidelity generation.

The denoising tasks at different timesteps are similar yet different. On the one hand, the denoising tasks at different timesteps are similar in the sense that the model takes a noisy image from the same space as input and performs a denoising task. Intuitively, sharing knowledge between these tasks might facilitate more efficient training. Therefore, typical methods let the model take both the noisy image $x_t$ and the corresponding timestep $t$ as input, and share the model parameter across all timesteps. On the other hand, the denoising tasks at different timesteps have clear differences as the input noisy images are from different distributions, and the concrete "denoising" effect is also different. Li et al. (2023a) demonstrate that there is a substantial difference between the feature distributions in different timesteps. Fang et al. (2023b) show that the larger (noisy) timesteps tend to generate the low-frequency and the basic image content, while the smaller timesteps tend to generate the high-frequency and the image details.

We further study the conflicts of different timesteps during the training of the diffusion model. Fig. 1(a) shows the gradient similarity of different timesteps. We can observe that the diffusion models have *dissimilar gradients at different timesteps, especially the non-adjacent timesteps*, indicating a conflict between the optimization direction from different timesteps, as shown in Fig. 1(b). In one word, this gradient conflict indicates that different denoising tasks might have a negative interference with each other during training, which may harm the overall performance.

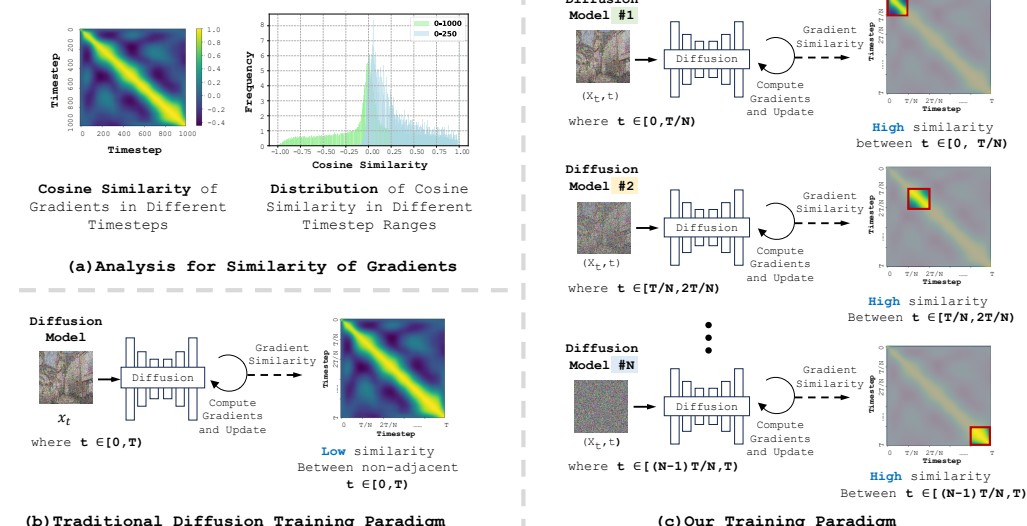

Figure 1: (a) Cosine similarity between gradients at different timesteps on CIFAR10 & distribution of gradients similarity in $t \in [0, 1000]$ and $t \in [0, 250]$. Non-adjacent timesteps have low similarity, indicating conflicts during their training. In contrast, adjacent timesteps have similar gradients. (b) & (c): Comparison between the traditional and our training paradigm: The previous paradigm trains one diffusion model on all timesteps, leading to conflicts in different timesteps. Our method addresses this problem by decoupling the training of diffusion models in $N$ different timestep ranges.

Considering the similarity as well as difference of these denoising tasks, the next natural and crucial question is "*how can we promote effective knowledge sharing as well as avoid negative interference between multiple denoising tasks?*". Timestep-wise model ensemble (Liu et al., 2023; Balaji et al., 2022) solves this problem by training and inferring multiple different diffusion models at various timesteps to avoid negative interference, though introducing huge additional storage and memory overhead. For instance, Liu et al. (2023) employs 6 diffusion models during inference, leading to around $6\times$ increase in storage and memory requirements, which renders the method impractical in application. Additionally, various loss reweighting strategies (Salimans & Ho, 2022; Hang et al., 2023) solve this problem by balancing different denoising tasks and mitigating negative interference. However, it may alleviate but can not truly solve the gradient conflicts in different timesteps.

In this work, considering the challenges faced by timestep-wise model ensemble and loss reweighting, we propose **De**couple-then-**Me**rge (**DeMe**), a novel finetuning framework for diffusion models that achieves the best side of both worlds: mitigated training interference across different denoising tasks and inference without extra overhead. DeMe begins with a pretrained diffusion model and then finetunes its separate versions tailored to no-overlapped timestep ranges to avoid the negative interference of gradient conflicts. Several training techniques are introduced during this stage to preserve the benefits of knowledge sharing in different timesteps. Then, the post-finetuned diffusion models are merged into a single model in their parameter space, enabling effective knowledge sharing across multiple denoising tasks.

Specifically, as shown in Fig. 1(c), we divide the overall timestep range $[0, T)$ into multiple adjacent timestep ranges with no overlap as $\{[(i-1)T/N, iT/N)\}_{i=1}^{N}$, where $T$ denotes the maximal timestep and $N$ denotes the number of timestep ranges. Then, we finetune a pretrained diffusion model for each timestep range by only training it with the timesteps inside this range. As a result, we decouple the training of diffusion models at different timesteps. The gradients of different timesteps will not be accumulated together and their conflicts are naturally avoided. Besides, as shown in Fig. 2, we further introduce the following three simple but effective techniques during the finetuning stage, including *Consistency Loss* and *Probabilistic Sampling* to preserve the benefits from knowledge sharing across different timesteps, and *Channel-wise Projection* that directly enables the model to learn the channel-wise difference in different timesteps.

After the finetuning stage, we obtain $N$ diffusion models learned the knowledge in $N$ different timesteps ranges, which also lead to $N$ times costs in storage and memory. Then, we eliminate the additional costs by merging all these $N$ models into a single model in their parameter space with the model merging technique (Ilharco et al., 2022). In this way, the obtained merged model has the same

computation and parameter costs as the original diffusion model while maintaining the knowledge from the $N$ finetuned model, which indicates a notable improvement in generation quality.

Extensive experiments on 6 datasets have verified the effectiveness of DeMe for both unconditional and text-to-image generation. In summary, our contributions can be summarized as follows.

- We propose to decouple the training of diffusion models by finetuning multiple diffusion models in different timestep ranges. Three simple but effective training techniques are introduced to promote knowledge sharing between multiple denoising tasks in this stage.

- We propose to merge multiple finetuned diffusion models, each specialized for different timestep ranges, into a single diffusion model, which significantly enhances generation quality without any additional costs in computation, storage, and memory access. To the best of our knowledge, we are the first to merge diffusion models across different timesteps.

- Abundant experiments have been conducted on six datasets for both unconditional and text-to-image generation, demonstrating significant improvements in generation quality.

We note that our framework of combining task-specific training with parameter-space merging offers a novel method for multi-task learning, distinct from existing loss-balancing techniques (Kendall et al., 2018; Sener & Koltun, 2018), and can be potentially extended to general multi-task scenarios.

## 2 RELATED WORK

**Diffusion Models.** Diffusion Probabilistic Models(DPMs) (Ho et al., 2020; Nichol & Dhariwal, 2021; Dhariwal & Nichol, 2021; Sohl-Dickstein et al., 2015; Song et al., 2020b) represent a family of generative models that generate samples via a progressive denoising mechanism, starting from a random Gaussian distribution. Given that diffusion models suffer from slow generation and heavy computational costs, previous works have focused on improving diffusion models in various aspects, including model architectures (Peebles & Xie, 2023; Rombach et al., 2022b), faster sampler (Song et al., 2020a; Lu et al., 2022; Liu et al., 2022), prediction type and loss weighting (Hang et al., 2023; Salimans & Ho, 2022). Besides, a few works have attempted to accelerate DPMs generation through pruning (Fang et al., 2023a), quantization (Shang et al., 2023; Li et al., 2023b) and knowledge distillation (Kim et al., 2023; Salimans & Ho, 2022; Luhman & Luhman, 2021; Meng et al., 2023), which have achieved significant improvement on the generation efficiency. Motivated by the excellent generative capacity of diffusion models, DPMs have been developed in several applications, including text-to-image generation (Rombach et al., 2022b; Ramesh et al., 2022; Saharia et al., 2022b), video generation (Ho et al., 2022; Blattmann et al., 2023), image restoration (Saharia et al., 2022c), natural language generation (Li et al., 2022), audio synthesis (Kong et al., 2020), 3D content generation (Poole et al., 2022), ai4science such as protein structure generation (Wu et al., 2024), among others.

**Training of Diffusion Models & Multi-task Learning.** Multi-task Learning (MTL) is aimed at improving generalization performance by leveraging shared information across related tasks. The objective of MTL is to learn multiple related tasks jointly, allowing models to generalize better by learning representations that are useful for numerous tasks (Crawshaw, 2020). Despite its success in various applications, MTL faces significant challenges, particularly negative transfer (Wang et al., 2020; Crawshaw, 2020), which can degrade the performance of individual tasks when jointly trained. The training paradigm of diffusion models could be viewed as a multi-task learning problem: diffusion models are trained by learning a sequence of models that reverse each step of noise corruption across different noise levels. A parameter-shared denoiser is trained on different noise levels concurrently, which may cause performance degradation due to negative transfer—a phenomenon where learning multiple denoising tasks jointly hinders performance due to conflicts in timestep-specific denoising information. To better balance the learning of denoising tasks across different noise levels, previous works reweight training loss on different timesteps, improving diffusion model performance (Ho et al., 2020; Salimans & Ho, 2022) or accelerating training convergence (Hang et al., 2023). Go et al. (2024) analyze and improve the diffusion model by exploring task clustering and applying various MTL methods to diffusion model training. Kim et al. (2024) analyze the difficulty of denoising tasks and propose a novel easy-to-hard learning scheme for progressively training diffusion models. Different from (Go et al., 2024), we analyze gradient conflicts between timesteps,

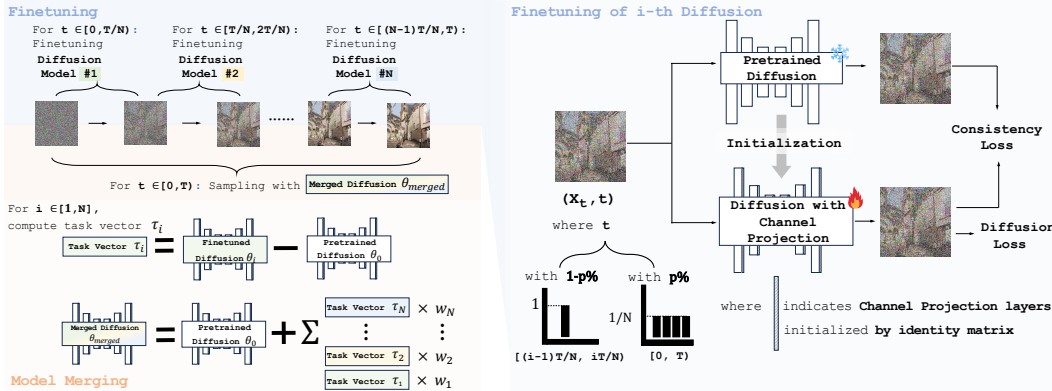

Figure 2: Pipeline of our framework. The following training techniques are incorporated into the finetuning process: **Consistency loss** preserves the original knowledge of diffusion models learned at all timesteps by minimizing the difference between pre-finetuned and post-finetuned diffusion models. **Probabilistic sampling** strategy samples from both the corresponding and other timesteps with different probabilities, helping the diffusion model overcome forgetting knowledge from other timesteps. **Channel-wise projection** enables the diffusion model to directly capture the feature difference in channel dimension. Model merging scheme merges the parameters of all the finetuned models into one unified model to promote the knowledge sharing across different timestep ranges.

propose to decouple the training of diffusion models by finetuning multiple diffusion models in different timestep ranges, and merge these models in the parameter space.

## 3 METHODOLOGY

### 3.1 PRELIMINARY

The fundamental concept of diffusion models is to generate images by progressively applying denoising steps, starting from random Gaussian noise $x_T$, and gradually transforming it into a structured image $x_0$. Diffusion models consist of two phases: the forward process and the reverse process. In the forward process, a data point $\mathbf{x}_0 \sim q(\mathbf{x})$ is randomly sampled from the real data distribution, then gradually corrupted by adding noise step-by-step $q(x_t \mid x_{t-1}) = \mathcal{N}(x_t; \sqrt{1 - \beta_t} x_{t-1}, \beta_t I)$, where $t$ is the current timestep and $\beta_t$ is a pre-defined variance schedule that schedules the noise. In the reverse process, diffusion models transform a random Gaussian noise $x_T \sim \mathcal{N}(0, \mathbf{I})$ into the target distribution by modeling conditional probability $q(x_{t-1} \mid x_t)$, which denoises the latent $x_t$ to get $x_{t-1}$. Formally, the conditional probability in the reverse process can be modeled as:

$$p_\theta(x_{t-1} \mid x_t) = \mathcal{N}\left(x_{t-1}; \frac{1}{\sqrt{\alpha_t}}\left(x_t - \frac{1 - \alpha_t}{\sqrt{1 - \bar{\alpha}_t}}\epsilon_\theta(x_t, t)\right), \beta_t\mathbf{I}\right), \tag{1}$$

where $\alpha_t = 1 - \beta_t$, $\bar{\alpha}_t = \prod_{i=1}^{T} \alpha_i$. $\epsilon_\theta$ denotes a noise predictor, which is usually an U-Net (Ronneberger et al., 2015) autoencoder in diffusion models, with current timestep $t$ and previous latent $x_t$ as input. It is usually trained with the objective function:

$$\mathcal{L}_\theta = \mathbb{E}_{t \sim U[0,T], x_0 \sim q(x), \epsilon \sim \mathcal{N}(0,1)}\left[\|\epsilon - \epsilon_\theta(x_t, t)\|^2\right], \tag{2}$$

where $T$ denotes the number of timesteps and $U$ denotes a uniform distribution. After training, a clean image $x_0$ can be obtained via an iterative denoising process from the random Gaussian noise $x_T \sim \mathcal{N}(0, \mathbf{I})$ with the modeled distribution $x_{t-1} \sim p_\theta(x_{t-1} \mid x_t)$ in Equation 1.

### 3.2 DECOUPLE THE TRAINING OF DIFFUSION MODEL

In this section, we demonstrate how to decouple the training of diffusion model. As illustrated in Fig. 1(c), we first divide the timesteps of $[0, T)$ into $N$ multiple continuous and non-overlapped timesteps ranges, which can be formulated as $\{[(i-1)T/N, iT/N)\}_{i=1}^{N}$. Subsequently, based on a

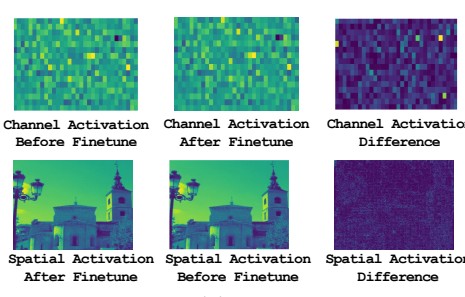
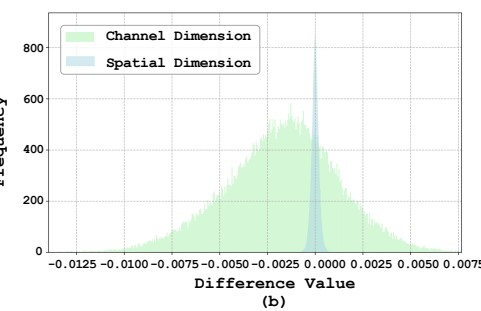

Figure 3: Visualization of the difference between the pre-finetuned and the post-finetuned diffusion model on the channel and spatial dimensions. (a) Visualization of channel activation, spatial activation, and their difference between the pre-finetuned and the post-finetuned model. (b) Distribution of difference for channel activation and spatial activation values. It can be observed that activation values **vary mostly in channel dimensions** during finetuning on a subset of timesteps.

diffusion model pretrained by Equation 1, we finetune a group of $N$ diffusion models $\{\epsilon_{\theta_i}\}_{i=1}^{N}$ on each of the $N$ timestep ranges. The training objective of $\epsilon_{\theta_i}$ which can be formulated as

$$\mathbb{E}_{t \sim U[(i-1)T/N, iT/N], x_0 \sim q(x), \epsilon \sim \mathcal{N}(0,1)} \left[ \| \epsilon - \epsilon_{\theta_i}(x_t, t) \|^2 \right]. \tag{3}$$

However, although Equation 3 can decouple the training of the diffusion model in different timesteps and avoid the negative interference between multiple denoising tasks, it also eliminates the positive benefits of learning from different timesteps, which may make the finetuned diffusion model overfit a specific timestep range and lose its knowledge in the other timesteps. Besides, it is also challenging for the diffusion model to capture the difference in different timesteps during finetuning. To address these problems, as shown in Fig. 2, we further introduce the following three techniques.

**Consistency Loss.** A consistency loss is introduced into the training process to minimize the difference between the pre-finetuned and post-finetuned diffusion model, which can be formulated as

$$\mathbb{E}_{t \sim U[(i-1)T/N, iT/N]} \left[ \| \epsilon_\theta(x_t, t) - \epsilon_{\theta_i}(x_t, t) \|^2 \right], \tag{4}$$

where $\epsilon_\theta(x_t, t)$ denotes the output of the original diffusion model. $\epsilon_{\theta_i}(x_t, t)$ denotes the output of $i_{th}$ post-finetuned diffusion model. Minimizing the consistency loss preserves the initial knowledge of the diffusion model, and ensures that the finetuned diffusion models do not differ significantly from the pre-finetuned diffusion model. Besides, the consistency loss also enhances the stability of the training process for finetuning diffusion models in the timestep range. Combining Equation 3 and Equation 4, we can derive the overall loss:

$$\mathbb{E}_{t \sim U[(i-1)T/N, iT/N], x_0 \sim q(x), \epsilon \sim \mathcal{N}(0,1)} \left[ \| \epsilon - \epsilon_{\theta_i}(x_t, t) \|^2 + \| \epsilon_\theta(x_t, t) - \epsilon_{\theta_i}(x_t, t) \|^2 \right]. \tag{5}$$

**Probabilistic Sampling.** To further preserve the initial knowledge learned at all the timesteps, we design a *Probabilistic Sampling* strategy which enables the finetuned model to mainly learn from its corresponding timestep range, but still possible to preserve the knowledge in the other timestep ranges. Concretely, during the finetuning of $i_{th}$ diffusion model, we sample $t$ from the timestep range $[(i-1)T/N, iT/N)$ with a probability of $1-p$, while sampling from the overall range $[0, T)$ with a probability $p$. The overall sampling strategy can be expressed as follows:

$$t \sim \begin{cases} [(i-1)T/N, iT/N), i \in [1, N] & \text{with probability } 1-p, \\ [0, T) & \text{with probability } p. \end{cases} \tag{6}$$

**Channel-wise Projection.** Fig. 3 shows the difference between the pre-finetuned and the post-finetuned diffusion models, demonstrating that there is a significant difference in the channel dimension instead of the spatial dimension, which further implies that the knowledge learned during finetuning in a timestep range is primarily captured by channel-wise mapping instead of spatial mapping. Based on this observation, we further apply a *channel-wise projection layer* to facilitate the training process by directly formulating the channel-wise mapping. Let $\mathbf{F}_t \in \mathbb{R}^{C \times H \times W}$ denote the intermediate feature map of the noise predictor $\epsilon_\theta(x_t, t)$ at the timestep $t$, where $C, H, W$ denote the number of channels, height, and width of the feature map $\mathbf{F}_t$, respectively. The channel-wise projection is designed as $\mathbb{P}(\mathbf{F_t}) = \mathbf{W} \cdot \mathbf{F}_t$, where $\mathbf{W} \in \mathbb{R}^{C \times C}$ is a learnable projection matrix that enables the diffusion model to directly capture the feature difference in the channel dimension. Please note that we initialize $\mathbf{W}$ as an identity matrix to stabilize the training process.

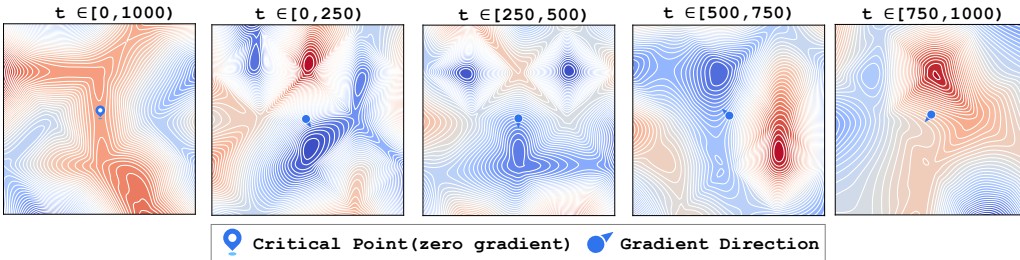

Figure 4: Loss landscape of the pretrained diffusion model in different timestep ranges on CIFAR10. Contour line density reflects the frequency of loss variations (*i.e.*, gradients), with blue representing low loss and red representing high loss. The pretrained model resides at the critical point (with zero gradients) with sparse contour lines for the overall timesteps $t \in [0, 1000)$, but when the training process is decoupled, it tends to be located in regions with densely packed contour lines, suggesting that there still exists gradients that enable pretrained model to escape from the critical point.

### 3.3 MERGING MODELS IN DIFFERENT TIMESTEP RANGES

After finetuning $N$ diffusion models in their corresponding timesteps, it is a natural step to ensemble these finetuned diffusion models in the inference stage. The sampling process under timestep-wise model ensemble scheme is achieved by inferring each post-finetuned diffusion model in its corresponding timestep range, which can be formulated as

$$p_\theta(x_{t-1} \mid x_t) = \mathcal{N}\left(x_{t-1}; \frac{1}{\sqrt{\alpha_t}}\left(x_t - \frac{1 - \alpha_t}{\sqrt{1 - \bar{\alpha}_t}}\epsilon_{\theta_i}(x_t, t)\right), \beta_t \mathbf{I}\right), i = \lfloor \frac{t \times N}{T} \rfloor. \quad (7)$$

For instance, the $i_{th}$ finetuned diffusion model is only utilized in timestep $t \in [(i-1)T/N, T/N)$. This inference scheme does not introduce additional computation costs during the inference period but does incur additional storage costs. Since model merging methods (Ilharco et al., 2022; Wortsman et al., 2022) can integrate diverse knowledge from each model, we propose to merge multiple finetuned diffusion models into *a single diffusion model*. Notably, this approach avoids additional computation or storage costs during inference while significantly improving generation quality.

**Model Merging Scheme.** Fig. 2 shows the overview of the model merge scheme. Inspired by model merging methods (Ilharco et al., 2022; Wortsman et al., 2022) that aim to merge the parameters of models finetuned in different datasets and tasks, we propose to merge multiple post-finetuned diffusion models. Specifically, we first compute the *task vectors* of different post-finetuned diffusion models, which indicates the difference in their parameters compared with the pre-finetuned version. The task vector $\tau_i$ of the $i_{th}$ finetuned diffusion model can be denoted as $\tau_i = \theta_i - \theta$, where $\theta$ and $\theta_i$ denote the parameters of the pre-finetuned and the $i_{th}$ post-finetuned diffusion model. Following previous work (Ilharco et al., 2022), the model merging can be achieved by adding all the task vectors to the pre-finetuned model, which can be formulated as

$$\theta_{\text{merged}} = \theta + \sum_{i=1}^{N} w_i \tau_i, \quad \text{where} \quad \tau_i = \theta_i - \theta, \quad (8)$$

where $w_i$ means merging weights of task vectors. To find the optimal combination of $w_i$, we use the grid search algorithm to explore this combination. In this scheme, we finally obtain $\theta_{\text{merged}}$ which can be applied across all timesteps in $[0, T)$, following the same inference process as in traditional diffusion models. As a result, the model merge scheme also leads to significant enhancement in generation quality without introducing any additional costs in computation or storage during inference.

**Proposition 1 (DeMe facilities escaping from critical point, informal Prop. 1. Proof in A.3)**
*Given a pretrained diffusion model with parameter $\theta$ converged in a critical point over the entire timesteps, which implies $\mathbb{E}_{t \sim U[0,T)}[\nabla \mathcal{L}_\theta(t)] = \mathbf{0}$. DeMe exhibits a non-zero gradient across the overall timesteps under the decouple-then-merge framework, indicating $\mathbb{E}_{t \sim U[0,T)}[\nabla \mathcal{L}_\theta^{\text{merge}}] \neq \mathbf{0}$, where $\mathcal{L}_\theta^{\text{merge}}$ is the objective function modeled by DeMe.*

Proposition 1 indicates that for a pretrained diffusion model converging in a critical point, DeMe enables it to escape from the critical point, allowing for further optimization, which demonstrates the effectiveness of DeMe. Proposition 1 can also be validated by the visualization results in Fig. 4.

Table 1: Quantitative results (FID, lower is better) on CIFAR10, LSUN-Church, and LSUN-Bedroom with DDPM. Numbers in the brackets indicate the FID difference compared with DDPM.

| Method | CIFAR10 | LSUN-Church | LSUN-Bedroom | #Iterations |
|---|---|---|---|---|
| Before-finetuning (Ho et al., 2020) | 4.42 | 10.69 | 6.46 | - |
| SNR+1 (Salimans & Ho, 2022) | 5.41 | 10.80 | 6.41 | 80K |
| Trun-SNR (Salimans & Ho, 2022) | 4.49 | 10.81 | 6.42 | 80K |
| Min-SNR-$\gamma$ (Hang et al., 2023) | 5.77 | 10.82 | 6.41 | 80K |
| DeMe (Before Merge) | 3.79 $_{(-0.63)}$ | 9.57 $_{(-1.12)}$ | 5.87 $_{(-0.59)}$ | 20K×4 |
| DeMe (After Merge) | **3.51** $_{(-0.91)}$ | **7.27** $_{(-3.42)}$ | **5.84** $_{(-0.62)}$ | 20K×4 |

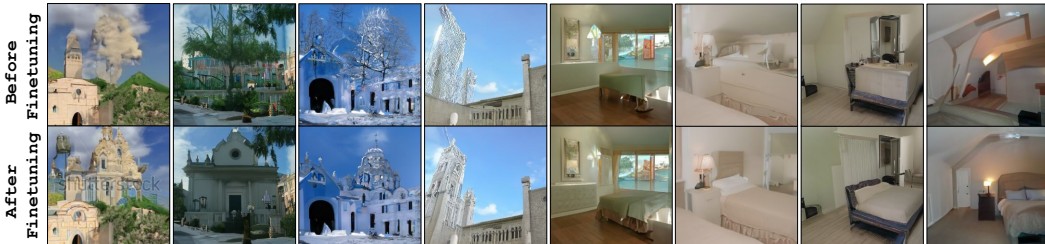

Figure 5: Qualitative comparison between our method and original DDPM on LSUN.

# 4 EXPERIMENTS

## 4.1 EXPERIMENT SETTING

**Datasets and Metrics.** For unconditional image generation datasets CIFAR10 (Krizhevsky et al., 2009), LSUN-Church, and LSUN-Bedroom (Yu et al., 2015), we generated 50K images for evaluation. For text-to-image generation, following the previous work (Kim et al., 2023), we finetune each model on a subset of LAION-Aesthetics V2 (L-Aes) 6.5+ (Schuhmann et al., 2022) and test model's capacity of zero-shot text-to-image generation on MS-COCO validation set (Lin et al., 2014), ImageNet1K (Deng et al., 2009) and PartiPrompts (Yu et al., 2022). Fréchet Inception Distance (FID) (Heusel et al., 2017) is used to evaluate the quality of generated images. CLIP score computed by CLIP-ViT-g/14 (Radford et al., 2021) is used to evaluate the text-image alignment.

**Baselines.** We choose three loss reweighting methods as baselines for comparison: *SNR+1*, *truncated SNR* (Salimans & Ho, 2022) and *Min-SNR-$\gamma$* (Hang et al., 2023). Appendix A.1 proves that the aforementioned diffusion loss weights can be unified under the same prediction target with different weight forms. Appendix A.2 demonstrates that our decouple-then-merge framework can also be formally transformed into the loss reweighting framework. To ensure a fair comparison, the baseline models are trained with an equal number of iterations with our training framework. Additionally, we also ensemble finetuned diffusion models and compare them with the merging scheme for a more detailed comparison. Please refer to the Appendix B for more implementation details.

## 4.2 QUANTITATIVE STUDY

**Results on Unconditional Generation.** Table 1 presents quantitative results on unconditional generation, demonstrating great improvement in generation quality across various unconditional image generation benchmarks. The model merging scheme achieves performance comparable to, or even better than, the ensemble scheme with a unified diffusion model, highlighting the superiority of the merging approach. Concretely, 0.63, 1.12, and 0.59 FID reduction can be observed on CIFAR10, LSUN-Church, and LSUN-Bedroom with the model ensemble scheme, respectively. The model merging scheme leads to 0.91, 3.42, and 0.62 FID reductions on CIFAR10, LSUN-Church, and LSUN-Bedroom, respectively. In contrast, previous loss weighting methods obtain very few FID reductions under the same finetuning setting and even harm the generation quality during finetuning.

**Results on Text-to-Image Generation.** Table 2 shows that DeMe outperforms the baselines in both image quality and text-image alignment, as demonstrated by the experiment results on text-to-image generation benchmarks for Stable Diffusion v1.4 (Rombach et al., 2022b). Specifically, on MS COCO, our ensemble method achieves a 0.64 FID reduction and a 0.03 CLIP score reduction, while merging method yields a 0.36 FID reduction along with a 0.23 CLIP score increase. On Ima-

Table 2: Quantitative studies on MS COCO, PartiPrompts and ImageNet with Stable Diffusion. Numbers in the brackets indicate the FID or CLIP Score difference compared with Stable Diffusion.

| Method | MS-COCO | | ImageNet | | PartiPrompts | #Iterations |
| --- | --- | --- | --- | --- | --- | --- |
| | FID↓ | CLIP Score↑ | FID↓ | CLIP Score↑ | CLIP Score↑ | |
| Before-finetuning (Rombach et al., 2022b) | 13.42 | 29.88 | 27.62 | 27.07 | 29.78 | - |
| SNR+1 (Salimans & Ho, 2022) | 13.92 | 29.96 | 27.56 | 27.03 | 29.86 | 80K |
| Trun-SNR (Salimans & Ho, 2022) | 13.93 | 29.95 | 27.60 | 27.05 | 29.85 | 80K |
| Min-SNR-$\gamma$ (Hang et al., 2023) | 13.92 | 29.93 | 27.59 | 27.02 | 29.87 | 80K |
| DeMe (Before Merge) | 12.78 $_{(-0.64)}$ | 29.85 $_{(-0.03)}$ | 26.36 $_{(-1.26)}$ | 26.90 $_{(-0.17)}$ | 30.02 $_{(+0.24)}$ | 20K×4 |
| DeMe (After Merge) | **13.06** $_{(-0.36)}$ | **30.11** $_{(+0.23)}$ | **27.23** $_{(-0.39)}$ | **27.09** $_{(+0.02)}$ | 29.98 $_{(+0.20)}$ | 20K×4 |

**Prompt I:** "*A graceful white horse galloping through a field of wildflowers, its mane flowing in the wind as the sun sets behind it.*"

*Before Finetuning: Loss of text-image alignment: as the sun sets behind it* 😣
*After Finetuning: Superb text-image alignment, lifelike horse* 🤩

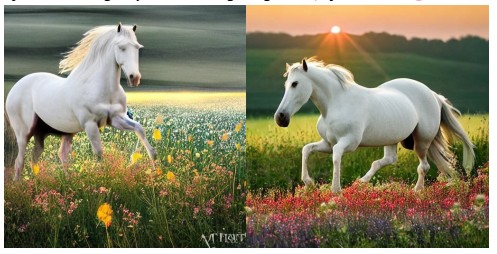

**Prompt II:** "*A tropical beach with crystal-clear turquoise water gently lapping against white sandy shores, tall palm trees swaying in the breeze under a clear blue sky.*"

*Before Finetuning: Loss of text-image alignment: white sandy shores* 😣
*After Finetuning: Superb text-image alignment, excellent coastal view* 🤩

**Prompt III:** "*A dolphin leaping out of the ocean in perfect harmony, water splashing around it as the sun sets on the horizon, casting a golden glow on the sea.*"

*Before Finetuning: Loss of text-image alignment: sun sets on the horizon..* 😣
*After Finetuning: Superb text-image alignment, photorealistic illustration* 🤩

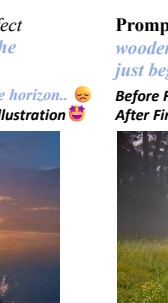
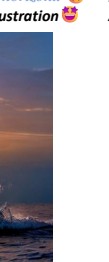

**Prompt IV:** "*A foggy morning in a quiet countryside, a small wooden cabin surrounded by wildflowers and tall grass, the sun just beginning to rise through the mist.*"

*Before Finetuning: Loss of text-image alignment: a small wooden cabin* 😣
*After Finetuning: Superb text-image alignment, serene landscape* 🤩

Before Finetuning     After Finetuning     Before Finetuning     After Finetuning

Figure 6: Qualitative comparison between our method and the original Stable Diffusion on various prompts. More qualitative results based on various text prompts could be found in Appendix G.2.

geNet1k, ensemble method results in a 1.26 FID reduction and a 0.17 CLIP score reduction, whereas merging method produces a 0.39 FID reduction and a 0.02 CLIP score increase. Additionally, on PartiPrompts, both the ensemble and merging schemes show improvements in CLIP score, with increases of 0.24 and 0.20, respectively. These results validate the effectiveness of DeMe, showing significant improvements in both image quality and text-image alignment.

## 4.3 QUALITATIVE STUDY

Fig. 5 depicts some synthesized images of LSUN and Fig. 6 depicts some fancy generated images given detailed prompts. As demonstrated in Fig. 5, our method has better **captured the underlying patterns in the images, specifically the church and bedroom scenes**, enabling a more detailed and accurate generation of the church and bedroom. While diffusion that before finetuning fails to generate churches or bedrooms, diffusion after finetuning successfully generates them with finer details. The finetuned diffusion demonstrates an improved ability to generate coherent and realistic representations of the target objects, as evidenced by the success in producing church-style buildings and bedroom-style interiors. Additional qualitative results on LSUN could be found in Appendix G.1.

Fig. 6 illustrates that our method **effectively generates images that align with the provided text descriptions**, resulting in generated images that are both more detailed and photorealistic. Prompts highlighted in bold indicate where Stable Diffusion fails to align the image with the text, whereas our method generates images with better text-image alignment. For example, in the middle image

pair of Fig. 6, Stable Diffusion fails to generate *a small wooden cabin* in image, while our method successfully captures the subject and preserves the detailed information described in the prompt. The finetuned Stable Diffusion model demonstrates an improved ability to generate visually coherent and contextually accurate images that closely match the nuances of the prompts, as highlighted in the comparison between before- and after-finetuning results, showcasing its enhanced capacity for text-to-image synthesis. More figures based on various text prompts could be found in Appendix G.2.

### 4.4 ABLATION STUDY

Our framework applies three training techniques to finetune diffusion model in different timesteps. As shown in Table 3, we conducted ablation studies on training techniques individually. All experiments are conducted on CIFAR10, with a 100-step DDIM sampler (Song et al., 2020a). Several key observations can be made: (i) The traditional training paradigm results in the poorest performance. With $N$ set to 1

Table 3: Ablation study on CIFAR10. $N$ denotes the number of finetuned models.

| $N$ | Probabilistic Sampling | Consistency Loss | Channel-wise Projection | FID ↓ |
|---|---|---|---|---|
| 1 | ✗ | ✗ | ✗ | 4.40 |
|   | ✗ | ✗ | ✔ | 4.45 |
| 8 | ✔ | ✗ | ✗ | 4.32 |
|   | ✔ | ✔ | ✗ | 4.27 |
|   | ✔ | ✔ | ✔ | 3.87 |

and none of the specialized training techniques applied-following the traditional diffusion training paradigm—the model yields a poor results, with a FID of 4.40. Gradient conflicts lead to negative interference across different denoising tasks, adversely affecting overall training. (ii) Channel-wise projection struggles to capture feature differences in the channel dimension without alleviating gradient conflicts. With $N$ set to 1 and Channel-wise projection applied, model yields a worse results, with a FID of 4.45. In contrast, with $N$ set to 8 and Channel-wise projection applied additionally, model yields the best results, with a FID of 3.87. We posit that channel-wise projection struggles to capture feature changes due to the significant differences across the timesteps. (iii) Dividing overall timesteps into $N$ non-overlapping ranges effectively alleviates gradient conflicts, resulting in a significant reduction in FID. For instance, with $N$ set to 8, introducing Probabilistic Sampling achieves a 0.08 FID reduction, while applying Consistency Loss yields a 0.13 FID reduction additionally. When all techniques are applied during finetuning, a total FID reduction of 0.53 is achieved. Our experimental results demonstrates that dividing overall timestep into non-overlapping ranges serves as a necessary condition. Building on this foundation, our training techniques significantly improve model performance. Sensitive studies on influence of $N$ and $p$ have been conducted in Appendix F, demonstrating that our method is robust to variations in the choices of $N$ and $p$.

## 5 DISCUSSION

**DeMe Enables Pretrained Model Escaping from the Critical Point.** In Prop. 1 we claim that DeMe facilities model moving away from the critical point, leading to further optimization. Fig. 4 presents some visualization results on the training loss landscape that supporting our claims. Two significant findings can be drawn from Fig. 4: (i) The pretrained diffusion model has converged when $t \in [0, 1000)$, residing at the critical point with sparse contour lines (*i.e.*, no gradient). However, it is evident that the pretrained model is not at an optimal point, as there are nearby points with lower training loss, suggesting a potential direction for further optimization. (ii) For different timestep ranges, the pretrained model tends to be situated in regions with densely packed contour lines (*i.e.*, larger gradient), suggesting that there exists an optimization direction. For instance, when $t \in [0, 250)$, the pretrained model stays at a point with frequent loss variations, indicating a potential direction for lower training loss. The decoupled training framework facilities the diffusion model to optimize more efficiently. Based on the above observation, DeMe decouples the training process, enabling the pretrained model to move away from the critical point, resulting in further improvement.

**Loss Landscape Visualization for Task Vectors.** To provide some intuitions, we visualize a two-dimensional training loss representation when applying two task vectors to merge finetuned models across various datasets, shown in Fig. 7(a). We utilize pretrained model $\theta$, two finetuned model $\theta_i(i = 1, 2)$ to obtain two task vectors $\tau_i(i = 1, 2)$, which span a plane in parameter space. We evaluate the diffusion training loss on this plane, and there are three key observations obtained from Fig. 7(a): (i) For both CIFAR10 and LSUN-Church, the training loss contours are basin-shaped and none of the model parameters are optimal, which means there exists a direction towards a better model parameters. (ii) The weighted sum of task vectors $\tau$ (*i.e.*, the interpolation of finetuned model

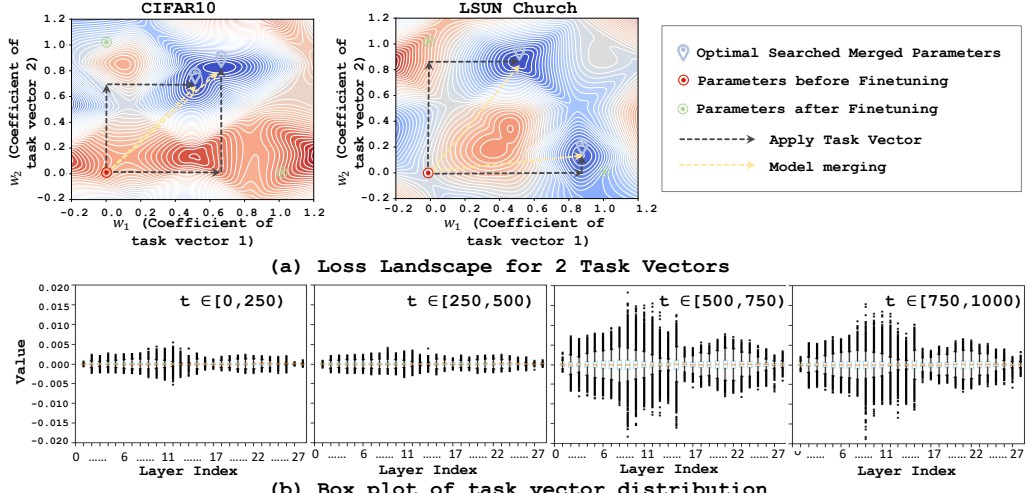

Figure 7: (a): Loss landscape for applying task vectors. The optimal model parameters are neither the pretrained one nor the finetuned one, but lie within the plane spanned by the task vectors computed in Sec. 3.3. We utilize the pretrained and two finetuned model parameters to obtain the two task vectors, respectively. Following (Wortsman et al., 2022; Garipov et al., 2018), we compute an orthonormal basis from the plane spanned by the task vectors. Axis denotes the movement direction in the parameter space. (b): Box plot of task vector distribution over different layers on LSUN-Church. Task vectors exhibit notable value in $t \in [500, 1000)$ but only slight value in $t \in [0, 500)$.

parameters $\theta_i$) can yield parameters with a lower training loss. For instance, on CIFAR10, the weighted sum of the task vectors can produce optimal model parameters, outperforming the two individual finetuned model parameters. (iii) The loss variation is relatively smooth, opening up the possibility to employ advanced search methods, such as evolutionary search, which could serve as a potential avenue for further improvement. In brief summary, the above observations indicate that by applying a weighted sum to the task vectors, a more optimal set of model parameters can be achieved, leading to a lower training loss.

**Task Vector Analysis.** DeMe employs the model merge technique to merge the multiple finetuned diffusion models by calculating the *linear combination of task vectors* introduced in Equation 8. Here we visualize the task vectors in Fig. 7(b), which shows significant differences between the task vectors in different timestep ranges. Specifically, the magnitude of task vectors has a larger value for $t \in [500, 1000)$ and a smaller value for $t \in [0, 500)$, indicating that there are more significant differences in parameters for diffusion models finetuned for $t \in [500, 1000)$. We suggest this because the original SNR loss term (Ho et al., 2020) has lower values in larger $t$. As a result, the original diffusion model bias to the gradients in smaller $t$ when larger $t$ and smaller $t$ have conflicts in gradients, leading to poor optimization for larger $t$. In contrast, DeMe decouples the training of diffusion models across larger $t$ and smaller $t$, allowing different timestep ranges to be optimized separately. Hence, the diffusion model finetuned on larger $t$ exhibits a more significant difference compared with the original model, which leads to better generalization quality.

# 6 CONCLUSION

Motivated by the observation that different timesteps in the diffusion model training have low similarity in their gradients, this paper proposes **DeMe**, which decouples the training of diffusion models in different timesteps and merge the finetuned diffusion models in parameter-space, thereby mitigating the negative impacts of gradient conflicts. Besides, three simple but effective training techniques have been introduced to facilitate the finetuning process, which preserve the benefits of knowledge sharing in different timesteps. Our experimental results on six datasets with both unconditional and text-to-image generation demonstrate that our approach leads to substantial improvements in generation quality without incurring additional computation or storage costs during sampling. The effectiveness of DeMe may promote more research work on the optimization of diffusion models. Additionally, the feasibility of combining task-specific training with parameter-space merging presented in this work may stimulate more research into diffusion model merging, and can be potentially extended to general multi-task learning scenarios.

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

# APPENDIX

## A  PROOF

In Sec. A.1 and Sec. A.2, we demonstrate that DeMe can be formally transformed into a loss reweighting framework, just like previous works (Salimans & Ho, 2022; Hang et al., 2023). Sec. A.3 is a proof for Prop. 1, which demonstrates the effectiveness of DeMe.

### A.1  THE DERIVATION OF SOME LOSS REWEIGHTING STRATEGIES

The standard diffusion loss can be formulated as follows:

$$\mathcal{L}_{\text{standard}} = \mathbb{E}_{t,x_0,\epsilon}\left[\|\epsilon - \epsilon_\theta(x_t,t)\|^2\right]. \tag{9}$$

It is worth noting that Equation 9 is identical to $\mathcal{L}_\theta$ in Equation 2. For the convenience of subsequent explanations, it has been restated here.

Actually, Equation 9 uses $\epsilon$ as the prediction target, but we can equivalently transform it into a loss function where $x_0$ is the prediction target:

$$\begin{aligned}
\mathcal{L}_{\text{standard}} &= \mathbb{E}_{t,x_0,\epsilon}\left[\|\epsilon - \epsilon_\theta(x_t,t)\|^2\right] \\
&= \mathbb{E}_{t,x_0,x_t}\left[\left\|\frac{1}{\sqrt{1-\bar{\alpha}_t}}\left(x_t - \sqrt{\bar{\alpha}_t}x_0\right) - \frac{1}{\sqrt{1-\bar{\alpha}_t}}\left(x_t - \sqrt{\bar{\alpha}_t}x_\theta(x_t,t)\right)\right\|^2\right] \\
&= \mathbb{E}_{t,x_0,x_t}\left[\frac{\bar{\alpha}_t}{1-\bar{\alpha}_t}\|x_0 - x_\theta(x_t,t)\|^2\right] \\
&= \mathbb{E}_{t,x_0,x_t}\left[\text{SNR}(t)\|x_0 - x_\theta(x_t,t)\|^2\right],
\end{aligned}$$

where $\text{SNR}(t) = \frac{\bar{\alpha}_t}{1-\bar{\alpha}_t}$. Salimans & Ho (2022) propose a loss reweighting strategy named Truncated SNR:

$$\mathcal{L}_{\text{Trun-SNR}} = \mathbb{E}_{t,x_0,x_t}\left[\max\left(\text{SNR}(t),1\right)\|x_0 - x_\theta(x_t,t)\|^2\right],$$

which is primarily designed to prevent the weight coefficient from reaching zero as the SNR approaches zero. Additionally, Salimans & Ho (2022) propose a new prediction target:

$$v = \sqrt{\bar{\alpha}_t}\epsilon - \sqrt{1-\bar{\alpha}_t}x_0. \tag{10}$$

Similarly, the objective function that uses $v$ as the prediction target can also be equivalently transformed into an objective function where $x_0$ is the prediction target:

$$\begin{aligned}
\mathcal{L}_{\text{SNR+1}} &= \mathbb{E}_{t,x_0,v}\left[\|v - v_\theta(x_t,t)\|^2\right] \\
&= \mathbb{E}_{t,x_0,x_t}\left[\left\|\left(\sqrt{\bar{\alpha}_t}\left(\frac{1}{\sqrt{1-\bar{\alpha}_t}}\left(x_t - \sqrt{\bar{\alpha}_t}x_0\right)\right) - \sqrt{1-\bar{\alpha}_t}x_0\right)\right.\right. \\
&\qquad\left.\left. - \left(\sqrt{\bar{\alpha}_t}\left(\frac{1}{\sqrt{1-\bar{\alpha}_t}}\left(x_t - \sqrt{\bar{\alpha}_t}x_\theta(x_t,t)\right)\right) - \sqrt{1-\bar{\alpha}_t}x_\theta(x_t,t)\right)\right\|^2\right] \\
&= \mathbb{E}_{t,x_0,x_t}\left[\frac{1}{1-\bar{\alpha}_t}\|x_0 - x_\theta(x_t,t)\|^2\right] \\
&= \mathbb{E}_{t,x_0,x_t}\left[(\text{SNR}(t)+1)\|x_0 - x_\theta(x_t,t)\|^2\right].
\end{aligned}$$

Furthermore, a new reweighting strategy (Hang et al., 2023) has been proposed to achieve accelerated convergence during the training process, named Min-SNR-$\gamma$:

$$\mathcal{L}_{\text{Min-SNR-}\gamma} = \mathbb{E}_{t,x_0,x_t}\left[\min\left(\text{SNR}(t),\gamma\right)\|x_0 - x_\theta(x_t,t)\|^2\right].$$

In a word, if the prediction target is $x_0$, the reweighting strategies can be written as follows:

- Standard diffusion loss (Ho et al., 2020):

$$\mathcal{L}_{\text{standard}} = \mathbb{E}_{t,x_0,x_t} \left[ \text{SNR}(t) \left\| x_0 - x_\theta(x_t,t) \right\|^2 \right] \tag{11}$$

- SNR+1 (Salimans & Ho, 2022):

$$\mathcal{L}_{\text{SNR+1}} = \mathbb{E}_{t,x_0,x_t} \left[ (\text{SNR}(t) + 1) \left\| x_0 - x_\theta(x_t,t) \right\|^2 \right] \tag{12}$$

- Truncated SNR (Salimans & Ho, 2022):

$$\mathcal{L}_{\text{Trun-SNR}} = \mathbb{E}_{t,x_0,x_t} \left[ \max\left(\text{SNR}(t), 1\right) \left\| x_0 - x_\theta(x_t,t) \right\|^2 \right] \tag{13}$$

- Min-SNR-$\gamma$ (Hang et al., 2023):

$$\mathcal{L}_{\text{Min-SNR-}\gamma} = \mathbb{E}_{t,x_0,x_t} \left[ \min\left(\text{SNR}(t), \gamma\right) \left\| x_0 - x_\theta(x_t,t) \right\|^2 \right] \tag{14}$$

## A.2 Transform DeMe Framework to Loss Reweighting Framework

In Sec. 3.2, we divide the overall timesteps $[0, T]$ into $N$ multiple continuous and non-overlapped timesteps ranges, which can be formulated as $\left\{ {}^{(i-1)T}/{N}, {}^{iT}/{N} \right\}_{i=1}^{N}$. For each range, we finetune a diffusion model $\epsilon_{\theta_i}$, the training objective of $\epsilon_{\theta_i}$ can be formulated as follows:

$$\mathcal{L}_i = \mathbb{E}_{t \sim U\left[\frac{(i-1)T}{N}, \frac{iT}{N}\right], x_0, \epsilon} \left[ \left\| \epsilon - \epsilon_{\theta_i}(x_t, t) \right\|^2 + \left\| \epsilon_\theta(x_t, t) - \epsilon_{\theta_i}(x_t, t) \right\|^2 \right]. \tag{15}$$

In Equation 15, the first term is the standard diffusion loss over the subrange, and the second term is the consistency loss, ensuring that the finetuned model $\epsilon_{\theta_i}$ stays close to the original model $\epsilon_\theta$.

In Sec. 3.3, we compute task vector $\tau_i = \theta_i - \theta$ after finetuning $\epsilon_{\theta_i}$, and merge $N$ post-finetuned diffusion models by

$$\theta_{\text{merged}} = \theta + \sum_{i=1}^{N} w_i \tau_i, \tag{16}$$

where $w_i$ are the merging weights determined(via grid search).

The update in parameters $\tau_i$ on due to finetuning on timestep range $i$ is:

$$\tau_i = \theta_i - \theta = -\eta \nabla_\theta L_i, \tag{17}$$

where $\eta$ is the learning rate. The merged model's parameters in Equation 17 could be rewritten as:

$$\theta_{\text{merged}} = \theta - \eta \sum_{i=1}^{N} w_i \nabla_\theta \mathcal{L}_i, \tag{18}$$

which implies $\theta_{\text{merged}}$ minimizes the combined loss

$$\mathcal{L}_{\text{merged}} = \sum_{i=1}^{N} w_i \mathcal{L}_i. \tag{19}$$

$L_i$ is computed over its respective timestep range, which menas $L_{\text{merged}}$ can be viewed as an integration over the entire timestep range with a piecewise constant weighting function $w(t)$. We rewrite $L_{\text{merged}}$ as:

$$\mathcal{L}_{\text{merged}} = \mathbb{E}_{t \sim U[0,T], x_0, \epsilon} \left[ w(t) \cdot \left\| \epsilon - \epsilon_\theta(x_t, t) \right\|^2 + \left\| \epsilon_\theta(x_t, t) - \epsilon_{\theta_i}(x_t, t) \right\|^2 \right], \tag{20}$$

where

$$w(t) = \begin{cases} w_i, & \text{if } t \in \left[\frac{(i-1)T}{N}, \frac{iT}{N}\right) \\ 0, & \text{otherwise} \end{cases}. \tag{21}$$

In Sec. 3.2, we propose to use $\theta$ to initialize $\theta_i$ and to utlize consistency loss to unsure $\epsilon_\theta(x_t, t) \approx \epsilon_{\theta_i}(x_t, t)$, which means that the second term in Equation 20 becomes negligible. The merged loss simplifies to

$$\mathcal{L}_{\text{merged}} = \mathbb{E}_{t \sim U[0,T], x_0, \epsilon} \left[ w(t) \cdot \| \epsilon - \epsilon_\theta(x_t, t) \|^2 \right] \tag{22}$$

$$= \mathbb{E}_{t \sim U[0,T], x_0, \epsilon} \left[ w(t) \cdot \text{SNR}(t) \cdot \| x_0 - x_\theta(x_t, t) \|^2 \right]. \tag{23}$$

Equation 23 is exactly the form of a reweighted loss function over timesteps, similar to Equations 11–14.

### A.3 PROOF OF PROP.1: DEME FACILITIES ESCAPING FROM CRITICAL POINT

Assuming that a pretrained diffusion model with parameter $\theta$ has converged over the entire timesteps $[0, T]$. Its gradient expectation across all the timesteps should be zero, indicating

$$\mathbb{E}_{t \sim [0,T)} [\nabla \mathcal{L}_\theta(t)] = \mathbf{0}. \tag{24}$$

Considering the complexity in non-convex optimization, Equation 24 also implies the possibility that the converged point $\theta$ may be a critical point. In this setting, we show that DeMe enables the diffusion model to escape from the critical point. The objective of DeMe can be formulated as

$$\mathcal{L}_\theta^{\text{merged}} = \sum_{i=1}^N w_i \mathcal{L}_{\theta,i}, \tag{25}$$

where $\mathcal{L}_{\theta,i}$ is the loss function of finetuning the $i_{th}$ diffusion models parameterized by $\theta$, which has been introduced in Equaltion 19. As an example with $N = 2$, DeMe decouples the training process into different timestep ranges $[0, T_1), [T_1, T)$, respectively. The expection of the gradients for training loss of DeMe can be formulated as

$$\mathbb{E}_{t \sim U[0,T)} \left[ \nabla \mathcal{L}_\theta^{\text{merged}} \right] = \mathbb{E}_{t \sim U[0,T)} \left[ w_1 \nabla \mathcal{L}_{\theta,1}(t) + w_2 \nabla \mathcal{L}_{\theta,2}(t) \right]. \tag{26}$$

DeMe decouples the training process into different timestep ranges by sampling timestep $t$ in their corresponding timesteps, which means

$$w_1 \mathbb{E}_{t \sim U[0,T)} [\nabla \mathcal{L}_{\theta,1}(t)] = w_1 \mathbb{E}_{t \sim U[0,T_1)} [\nabla \mathcal{L}_\theta(t)] \tag{27}$$

$$w_2 \mathbb{E}_{t \sim U[0,T)} [\nabla \mathcal{L}_{\theta,2}(t)] = w_2 \mathbb{E}_{t \sim U[T_1,T)} [\nabla \mathcal{L}_\theta(t)]. \tag{28}$$

Therefore, Equation 26 can be rewritten as follows:

$$\begin{aligned}
&\mathbb{E}_{t \sim U[0,T)} \left[ \nabla \mathcal{L}_\theta^{\text{merged}} \right] \\
&= w_1 \mathbb{E}_{t \sim U[0,T_1)} [\nabla \mathcal{L}_\theta(t)] + w_2 \mathbb{E}_{t \sim U[T_1,T)} [\nabla \mathcal{L}_\theta(t)] \\
&= w_1 \left[ \mathbb{E}_{t \sim U[0,T_1)} [\nabla \mathcal{L}_\theta(t)] + \mathbb{E}_{t \sim U[T_1,T)} [\nabla \mathcal{L}_\theta(t)] \right] + (w_2 - w_1) \mathbb{E}_{t \sim U[T_1,T)} [\nabla \mathcal{L}_\theta(t)].
\end{aligned} \tag{29}$$

According to Equation 24, we have

$$\mathbb{E}_{t \sim U[0,T_1)} [\nabla \mathcal{L}_\theta(t)] + \mathbb{E}_{t \sim U[T_1,T)} [\nabla \mathcal{L}_\theta(t)] = \mathbb{E}_{t \sim [0,T)} [\nabla \mathcal{L}_\theta(t)] = \mathbf{0}. \tag{30}$$

Besides, because $w_1 \neq w_2$, $\mathbb{E}_{t \sim U[T_1,T)} [\nabla \mathcal{L}_\theta(t)] \neq \mathbf{0}$, we can derive:

$$\mathbb{E}_{t \sim U[0,T)} \left[ \nabla \mathcal{L}_\theta^{\text{merged}} \right] = \mathbb{E}_{t \sim U[0,T)} [w_1 \nabla \mathcal{L}_{\theta,1}(t) + w_2 \nabla \mathcal{L}_{\theta,2}(t)] \neq \mathbf{0}. \tag{31}$$

Equation 31 proves that DeMe exhibits a non-zero gradient in the critical point where the original diffusion converged, which may provide an explanation for the effectiveness of DeMe.

## B EXPERIMENTAL DETAILS

**Implementation Details.**  During the finetuning process, we set $N = 4$ for all four datasets, $p = 0.4$ for CIFAR10 dataset, $p = 0.3$ for LSUN-Church, LSUN-Bedroom, and L-Aes 6.5+ datasets. For CIFAR10, each model is trained for 20K iterations with a batch size of 64 and a learning rate of 2e-4. For LSUN-Church, LSUN-Bedroom, and L-Aes 6.5+ datasets, each model is trained for 20K iterations with a batch size of 16, a learning rate of 5e-5, and gradient accumulation is set to 4. We employ a 50-step DDIM sampler (Song et al., 2020a) for DDPM and a 50-step PNDM sampler (Liu et al., 2022) for Stable Diffusion. All experiments are implemented on NVIDIA A100 80GB PCIe GPU and NVIDIA GeForce RTX 4090.

**Dataset Details.** For unconditional image generation datasets CIFAR10 (Krizhevsky et al., 2009), LSUN-Church, and LSUN-Bedroom (Yu et al., 2015), we generated 50K images to obtain the Fréchet Inception Distance (FID) (Heusel et al., 2017) for evaluation. For zero-shot text-to-image generation, following the previous work (Kim et al., 2023), we finetune each model on a subset of LAION-Aesthetics V2 (L-Aes) 6.5+ (Schuhmann et al., 2022), containing 0.22M image-text pairs. We use 30K prompts from the MS-COCO validation set (Lin et al., 2014), downsample the 512×512 generated images to 256×256, and compare the generated results with the whole validation set. We also use class names from ImageNet1K (Deng et al., 2009) and 1.6K prompts from PartiPrompts (Yu et al., 2022) to generate 2K images (2 images per class for ImageNet1k) and 1.6K images, individually. Fréchet Inception Distance (FID) (Heusel et al., 2017) and CLIP score (Radford et al., 2021) are used to evaluate the quality of generated images.

## C    RELATED WORKS ON MODEL MERGING

Merging models in parameter space emerged as a trending research field in recent years, aiming at enhancing performance on a single target task via merging multiple task-specific models (Matena & Raffel, 2022; Wortsman et al., 2022; Jin et al., 2022; Yang et al., 2024). In contrast to multi-task learning, model merging fuses model parameters by performing arithmetic operations directly in the parameter space (Ilharco et al., 2022; Yadav et al., 2024), allowing the merged model to retain task-specific knowledge from various tasks. Diffusion Soup (Biggs et al., 2024) suggests the feasibility of model merging in diffusion models by linearly merging diffusion models that are finetuned on different datasets, leading to a mixed-style text-to-image zero-shot generation. MaxFusion (Nair et al., 2024) fuses multiple diffusion models by merging intermediate features given the same input noisy image. LCSC (Liu et al., 2024) searches the optimal linear combination for a set of checkpoints in the training process, leading to a considerable training speedups and FID reduction. Unlike Diffusion Soup (Biggs et al., 2024), MaxFusion (Nair et al., 2024) and LCSC (Liu et al., 2024), DeMe leverages model merging to *fuse models finetuned at different timesteps*, combines the knowledge acquired at different timesteps, and resulting in improved model performance.

## D    RELATED WORKS ON TIMESTEP-WISE MODEL ENSEMBLE

Previous works (Balaji et al., 2022; Liu et al., 2023; Lee et al., 2024) have revealed that the performance of diffusion models varies across different timesteps, suggesting that diffusion models may excel at certain timesteps while underperforming at others. Inspired by this observation, several works Balaji et al. (2022); Lee et al. (2024); Zhang et al. (2023) explore the idea of proposing an ensemble of diffusion experts, each specialized for different timesteps, to achieve better overall performance. MEME (Lee et al., 2024) propose a multi-architecture and multi-expert diffusion models, which assign distinct architectures to different time-step intervals based on the frequency characteristics observed during the diffusion process. Zhang et al. (2023) introduce a multi-stage framework and tailored multi-decoder architectures to enhance the efficiency of diffusion models. eDiff-I (Balaji et al., 2022) propose training an ensemble of expert denoisers, each specialized for different stages of the iterative text-to-image generation process. Spectral Diffusion (Yang et al., 2023b) can also be viewed as an ensemble of experts, each specialized in processing particular frequency components during the iterative image synthesis. Go et al. (2023) leverages multiple guidance models, each specialized in handling a specific noise range, called Multi-Experts Strategy. OMS-DPM (Liu et al., 2023) propose a predictor-based search algorithm that optimizes the model schedule given a set of pretrained diffusion models.

## E    SIMILARITY BETWEEN TASK VECTORS

In Fig. 8, we analyze the cosine similarity between task vectors across different timestep ranges to explore how multiple finetuned diffusion models can be merged into a unified diffusion model through additive combination. We observe that task vectors from different timestep ranges are generally close to orthogonal, with cosine similarities remaining low, often near zero. We speculate that this orthogonality facilitates the additive merging of multiple finetuned diffusion models into a unified model with minimal interference, allowing for effective combination without conflicting gradients between the different timestep ranges. For instance, timestep ranges $t \in [0, 250)$ and

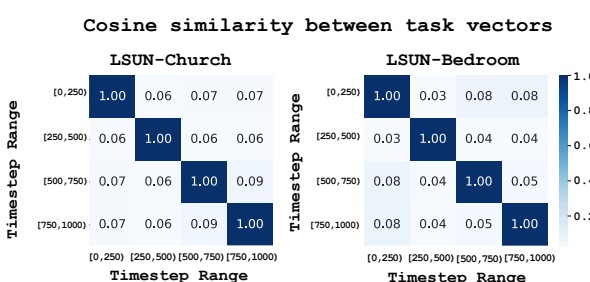

Figure 8: The cosine similarity between task vectors at different timestep ranges on two datasets: **Task vectors are nearly orthogonal between different timestep ranges**. This orthogonality suggests that knowledge from different timesteps is largely independent, allowing for effective additive combination of task vectors with minimal interference, thereby facilitating the merging of finetuned models.

$t \in [500, 750)$ on LSUN-Church exhibit a cosine similarity of 0.07, this relatively low value indicates that the task vectors for these two non-adjacent ranges are close to orthogonal, allowing for more effective combination during model merging with minimal interference between different denoising tasks.

Additionally, the slight deviations from orthogonality within different timestep ranges suggest some shared information between neighboring denoising tasks, reflecting a degree of continuity in the model's learning across these ranges. These deviations also highlight the effectiveness of the *Probabilistic Sampling Strategy* introduced in Sec. 3.2, which ensures a balance between specialization in the range and generalization across all timesteps, effectively preserving knowledge across different stages of denoising task training.

## F  SENSITIVE STUDY

DeMe decouples the training of diffusion models by finetuning multiple diffusion models in $N$ different timestep ranges. A larger $N$ indicates that the timesteps are divided into finer ranges, further reducing gradient conflicts and potentially enhancing the model's performance. Meanwhile, the probability $p$ determines the tradeoff between learning from specific and global timesteps, thereby influencing the model's performance. Therefore, we do some sensitive study on the influence of *number of ranges N* and *possibility p* on CIFAR10.

**Influence on Number of ranges $N$.** A larger $N$ implies each diffusion model is finetuned on a narrower timestep range, leading to less gradient conflicts. As illustrated in Fig. 9, it is observed that: (i) Training diffusion model across the entire timestep range results in the poorest performance. With $N = 1$, i.e., training diffusion on the overall timesteps, a minor improvement is achieved, with a FID of 4.34. We posit that severe gradient conflicts occurred, negatively impacting the overall training process. (ii) The finer the division of the overall timesteps into $N$ non-overlapping ranges, the more effectively it mitigates gradient conflicts, leading to a notable reduction in FID. For example, dividing the timesteps into 4 ranges can result in a 0.63 FID reduction, whereas dividing them into only 2 ranges leads to a reduction of just 0.4 FID. A larger N is associated with improved model performance, indicating reduced gradient conflicts. (iii) As $N$ increases, the model's training exhibits marginal utility. For instance, when $N$ exceeds 4, the FID no longer follows the decreasing trend observed when $N$ smaller than 4. This suggests that the model's gains notably diminish as $N$ increases. Considering the finetuning overhead and the complexity of model merging, we recommend $N = 4$ as a trade-off in practice.

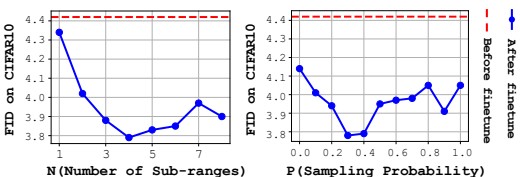

Figure 9: Sensitive study of the influence on the *number of ranges N* and *possibility p* of training of all timesteps on CIFAR10.

**Influence on Probability $p$.** Probability $p$ means a sampling probability $p$ for a diffusion model beyond its specific timespte range, which indicates a trade-off between specific knowledge and general knowledge. Varying choices of probability $p$ can enhance model performance to different extents. As shown in Fig. 9, it is observed that: (i) Training solely on either the full timestep range or specific

subranges limits knowledge sharing, resulting in only minor improvements. $P = 0$ corresponds to training across all timesteps, while $p = 1$ focuses exclusively on a specific range. Both of these settings restrict knowledge transfer between the overall and specific timestep ranges, leading to modest FID reductions of 0.28 and 0.37, respectively. (ii) Our method achieves varying degrees of improvement across the range $p \in [0, 1]$. When $p > 0.5$, sampling occurs more frequently over the overall timestep, while for $p < 0.3$, sampling is more concentrated in a specific timestep range. Both cases restrict knowledge transfer between the overall and specific timestep ranges, leading to minor FID improvements, shown in Fig. 9. To maximize the effectiveness of the method, we recommend using $p = 0.3$ or $p = 0.4$ in practice.

## G  ADDITIONAL QUALITATIVE EXPERIMENTS

### G.1  ADDITIONAL QUALITATIVE RESULTS ON LSUN FOR DDPM

In Fig. 10 and Fig. 11, additional qualitative results are presented. DeMe has more **effectively captured the underlying patterns in the images, specifically the church and bedroom scenes**, allowing for more detailed and accurate generation of these structures. The finetuned diffusion model demonstrates an improved ability to generate coherent and realistic representations of the target objects, as evidenced by the success in producing church-style buildings and bedroom-style interiors in the bottom rows of both figures.

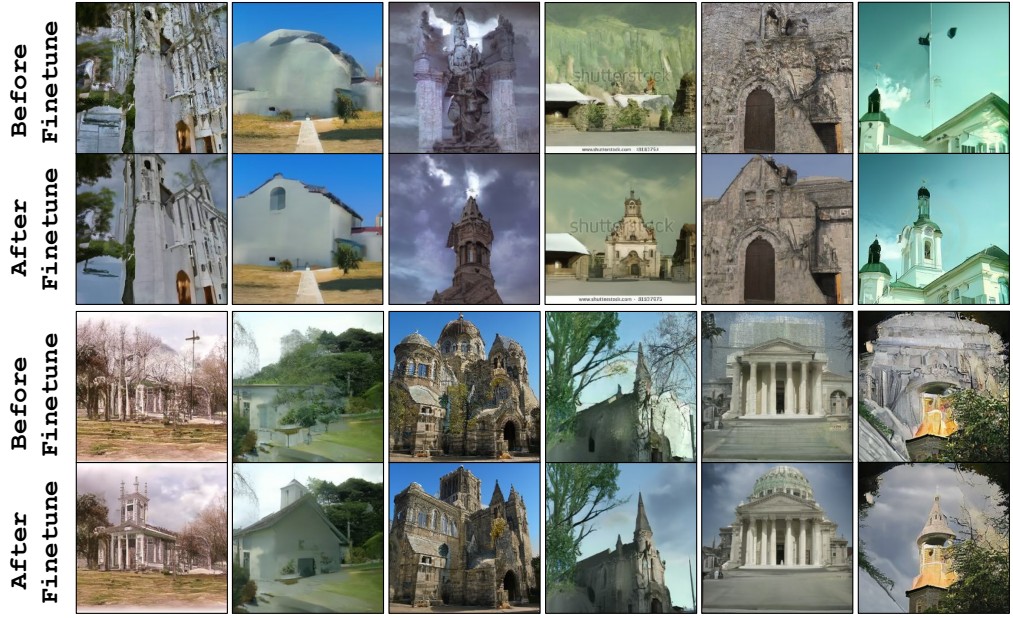

Figure 10: Additional qualitative results on LSUN-Church. The top row shows images generated by DDPM before finetuning, while the bottom row displays images generated by DDPM after finetuning using our training framework. In the bottom row, church-style buildings are successfully generated, whereas the top row fails to produce similar structures.

### G.2  ADDITIONAL QUALITATIVE RESULTS FOR STABLE DIFFUSION

In Fig. 12, Fig. 13 and Fig. 14, additional qualitative results are presented based on various detailed text prompts. DeMe more **effectively generates images that align with the provided text descriptions**, producing results that are both more detailed and photorealistic. The finetuned Stable Diffusion model demonstrates an improved ability to generate visually coherent and contextually accurate images that closely match the nuances of the prompts, as highlighted in the comparison between before- and after-finetuning results, showcasing its enhanced capacity for text-to-image synthesis.

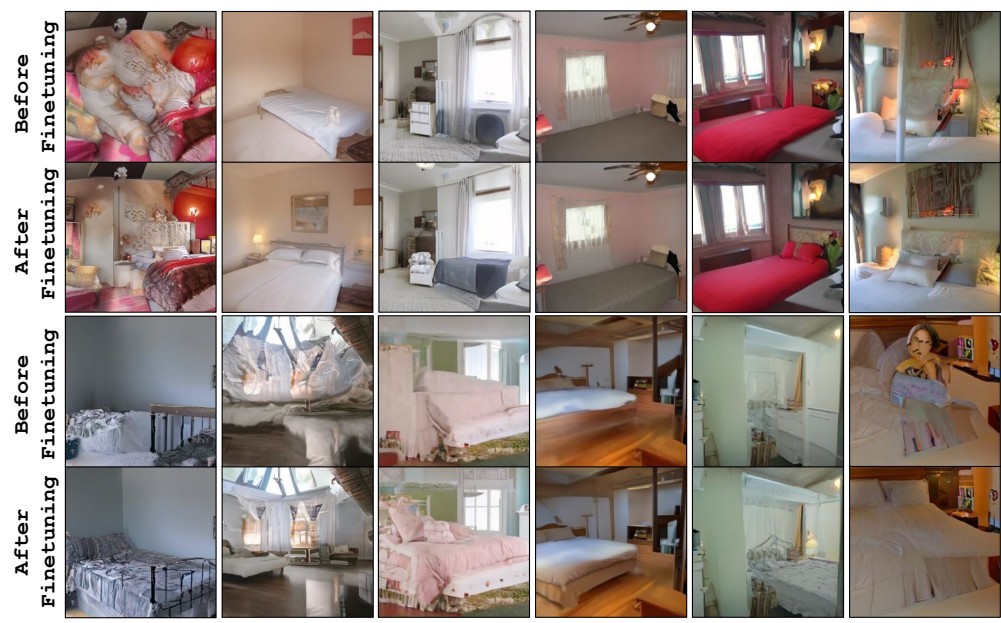

Figure 11: Additional qualitative results on LSUN-Bedroom. The top row shows images generated by DDPM before finetuning, while the bottom row displays images generated by DDPM after finetuning using our training framework. In the bottom row, bedroom scenes are successfully generated, whereas the top row fails to produce coherent structures.

**Prompt V:** "*A tranquil beach at sunrise, with soft waves lapping at the shore, palm trees swaying gently in the breeze, and a vibrant sky painted in shades of pink, orange, and purple.*"

*Before Finetuning: Loss of text-image alignment: palm trees swaying gently* 😣
*After Finetuning: Superb text-image alignment, gorgeous coastal view* 🤩

**Prompt VI:** "*A vast desert with rolling sand dunes, the golden sands stretching endlessly under a cloudless sky, a caravan of camels making its way across the horizon.*"

*Before Finetuning: Loss of text-image alignment: a caravan of camels* 😣
*After Finetune: Superb text-image alignment* 🤩

**Prompt VII:** "*A herd of wild horses galloping across a wide open field, their manes and tails flowing in the wind, dust kicking up beneath their hooves as they run.*"

*Before Finetuning: Loss of text-image alignment: dust kicking up..* 😣
*After Finetuning: Superb text-image alignment, vivid graphic depiction* 🤩

**Prompt VIII:** "*A scientist in a modern laboratory, wearing a white coat and safety goggles, examining a vial of glowing blue liquid with curiosity and focus.*"

*Before Finetuning: Loss details in scientist's body* 😣
*After Finetuning: Detailed generation in scientist's body* 🤩

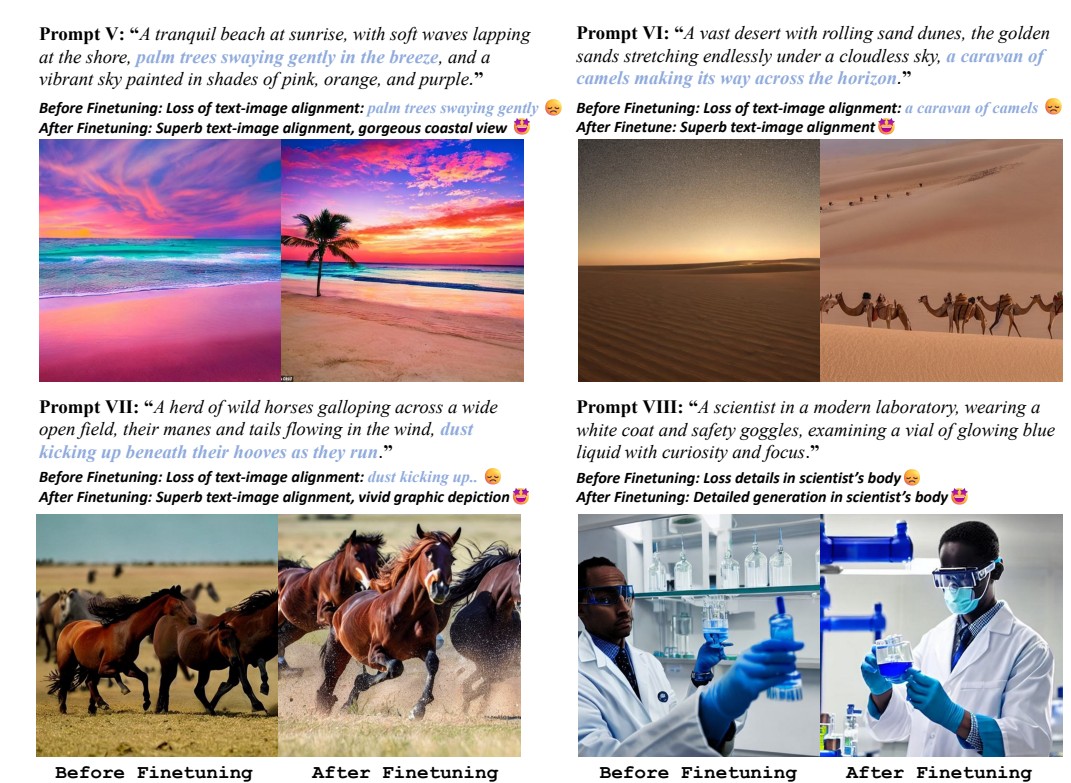

**Before Finetuning**  **After Finetuning**    **Before Finetuning**  **After Finetuning**

Figure 12: Additional qualitative results based on various text prompts for Stable Diffusion

**Prompt IX:** "*A majestic eagle soaring high above the mountains, its wings spread wide, gliding effortlessly through the sky with the distant peaks below.*"

*Before Finetuning: Loss of realistic graphic depiction* 😣
*After Finetuning: Superb text-image alignment, lifelike figure* 🤩

**Prompt X:** "*An old fisherman with weathered hands, sitting on a wooden dock by the sea, seagulls fly overhead in the golden light of sunset.*"

*Before Finetuning: Loss of text-image alignment: seagulls fly overhead* 😣
*After Finetuning: Superb text-image alignment* 🤩

**Prompt XI:** "*A dense, mystical forest in autumn, with rays of golden sunlight filtering through the vibrant orange and red leaves, a narrow path leading deeper into the trees.*"

*Before Finetuning: Loss of text-image alignment: sunlight filtering through..* 😣
*After Finetuning: Superb text-image alignment, photorealistic* 🤩

**Prompt XII:** "*A snowy winter landscape, a small village nestled in a valley, smoke rising from chimneys, snow-covered trees, and twinkling lights glowing warmly in the dusk.*"

*Before Finetuning : Loss of text-image alignment: smoke .. chimneys* 😣
*After Finetuning: Superb text-image alignment* 🤩

**Before Finetuning          After Finetuning                    Before Finetuning          After Finetuning**

Figure 13: Additional qualitative results based on various text prompts for Stable Diffusion

**Prompt XIII:** "*A breathtaking view of a waterfall cascading down into a lush jungle, with vibrant green foliage, moss-covered rocks, and a rainbow forming in the mist.*"

*Before Finetuning: Loss of text-image alignment: a rainbow* 😣
*After Finetuning: Superb text-image alignment* 🤩

**Prompt XIV:** "*A grand library with towering bookshelves filled with ancient tomes, a spiral staircase winding up to a high ceiling adorned with intricate frescoes, and soft light streaming through stained glass windows.*"

*Before Finetuning: Loss details: twisted staircase* 😣
*After Finetuning: Excellent details generation* 🤩

**Prompt XV:** "*A majestic lion standing proudly on a rocky ledge, its golden mane blowing in the wind, the vast African savannah stretching out behind it at sunset.*"

*Before Finetuning: Loss of text-image alignment: standing proudly..* 😣
*After Finetuning: Superb text-image alignment* 🤩

**Prompt XVI:** "*A curious red fox sitting in a snowy forest, its bright fur contrasting against the white snow, its ears perked up and eyes focused on something in the distance.*"

*Before Finetuning : Loss of text-image alignment: sitting* 😣
*After Finetuning: Superb text-image alignment* 🤩

**Before Finetuning          After Finetuning                    Before Finetuning          After Finetuning**

Figure 14: Additional qualitative results based on various text prompts for Stable Diffusion

