# OpenReview forum: "Decouple-Then-Merge: Towards Better Training for Diffusion Models"
_ICLR.cc/2025/Conference — ICLR 2025 Conference Withdrawn Submission_

### Official Review · Reviewer_MSPU · 2024-11-01

**Soundness:** 3
**Presentation:** 3
**Contribution:** 2
**Rating:** 5
**Confidence:** 3

**Summary:**

This paper proposes a method for enhancing generation quality by merging multiple fine-tuned diffusion models into one, with each model specialized in different timesteps. It introduces three techniques to improve generation results. Experimental results show that the proposed method achieves better performance compared to loss reweighting methods.

**Strengths:**

1. This paper proposes an effective method to enhance generation quality by merging multiple fine-tuned diffusion models into a single model.
2. In addition to generation results, this paper provides both empirical and theoretical analyses for better understanding.
3. The paper is well-written and easy to follow.

**Weaknesses:**

This paper is well-organized. However, I have some concerns as follows.

1. **Insufficient baselines**: As discussed in the paper (Section 1, 2 and appendix Section D), there are already multiple approaches that train multiple diffusion models at different timesteps, such as model ensemble methods and loss reweighting methods. However, only three loss reweighting methods are selected as baselines in the main results of Sections 4.2 and 4.3 (e.g., Table 1, line 355). Why were these specific methods chosen? Other timestep-wise model ensemble methods, including but not limited to eDiff-I and LCSC (see appendix section D), should also be considered for comparison.

2. **Motivation and analysis overlap with ensemble methods**: While this paper provides analysis supporting the proposed method, some may also apply to existing model ensemble methods, such as the motivation in Figure 1 (a)(b) and the analysis in Figure 4. Although the proposed method claims an advantage over ensemble methods by requiring only a single model for inference, additional comparisons and analyses between these methods would be beneficial.

3. **Choice of pretrained diffusion models**: The paper appears to use Stable Diffusion v1.4 for text-to-image generation (line 376). Since this version is relatively old (released in August 2022 [1]), it would be valuable to evaluate the proposed method on newer versions, such as Stable Diffusion 1.5, 2.1, XL, 3, or the recent 3.5.

4. **Missing related work**: The paper misses a relevant study[2], which proposes reweighting loss during training.

[1] https://stability.ai/news/stable-diffusion-public-release
[2] Choi, Jooyoung, et al. Perception Prioritized Training of Diffusion Models. CVPR 2022.

**Questions:**

Please refer to the weaknesses.

---

### Official Review · Reviewer_nCqS · 2024-11-02

**Soundness:** 2
**Presentation:** 3
**Contribution:** 2
**Rating:** 5
**Confidence:** 4

**Summary:**

This paper addresses a limitation in current diffusion models, which typically use a single model to handle all denoising steps, leading to gradient conflicts across timesteps and ultimately resulting in suboptimal generation quality. The authors analyze the gradients at various timesteps and demonstrate that these conflicts impact performance. To mitigate this, they propose fine-tuning multiple models, each specializing in a subset of the denoising steps. After fine-tuning, these models are combined, preserving computational and storage efficiency in the final model. The fine-tuning process incorporates three main components: a consistency loss, probabilistic sampling, and a channel projection layer. Empirical experiments confirm the effectiveness of this approach.

**Strengths:**

- The paper addresses a genuine limitation in current diffusion models.
- The paper provides a compelling and thorough analysis of the issue.
- The method’s effectiveness is supported by solid empirical results.
- The paper is well-written and easy to follow.

**Weaknesses:**

- While I agree with the paper’s main premise—that conflicting gradients across different timesteps can lead to suboptimal results—I find the chosen components of the method less than ideal for addressing this issue. The reliance on both consistency loss and probabilistic sampling suggests that this approach may serve as a partial fix rather than a comprehensive solution.
- The result of the activations mostly changing in the channel dimension is strange and it is unclear why this is the case.
- The improvements shown in the Stable Diffusion case are minimal, raising questions about the method’s overall impact.
- The method demonstrates effectiveness only in a fine-tuning context. It remains unclear if these improvements would hold if the diffusion model were pre-trained using this approach.

**Questions:**

**Major concerns and questions**

- Would the method work for pre-training, or is it effective only during fine-tuning? If limited to fine-tuning, what underlies this constraint? If applicable to pre-training as well, can the authors provide experimental results to support this?
- The paper would benefit from a more thorough explanation of why each component is necessary, as the need for these elements suggests a larger phenomenon at play. Specifically, the justification for the channel-wise projection feels vague and would benefit from clearer motivation.

**Other concerns and questions**

- In the Stable Diffusion experiments, were seeds fixed between the baseline and your method’s results? The outputs do not appear to come from the same seeds (in contrast to the results in Fig. 5 where they clearly come from the same seed). But I may be wrong. If the results do not come from the same seeds, the authors should revise the figure to display results with fixed seeds, otherwise this may be a very unfair cherry picking.
- An ablation study focusing solely on channel-wise projection with $N=8$, without probabilistic sampling or consistency loss, would be informative.
- Can the authors present results without the merging step, using each fine-tuned model sequentially to better understand the effect of merging?
- A visual comparison, with fixed seeds, of outputs at the end of each subset for the original model, the fine-tuned models, and the final merged model would help clarify the method’s contribution.

**Minor concerns and questions**

- I find the model merging section (lines 300–315) to be overly detailed. A simplified explanation, such as averaging $\Delta W_i$ (where $W$ represents pretrained weights and $W_i’ = W + \Delta W_i$ represents fine-tuned weights), could enhance clarity.
- In Eq. 5, are the terms balanced by a lambda parameter? If not, is there a trade-off in applying more or less of the consistency loss?
- In line 312, what does the grid search entail—what parameters are included in the grid? Also, in Eq. 8, what is $w_i$? As I understand, $T_i$ represents the task vector, but the role of $w_i$ remains unclear.
- Please elaborate on how the plot in Fig. 4 was generated. Does it reflect real data, and what is the specific setting?
___
In summary, the paper addresses a relevant issue, and its analysis highlights a real phenomenon. While the empirical results indicate performance improvements, the method does not seem to directly resolve the core problem. More intuition or justification for the different components would strengthen the paper, along with a few additional experiments to enhance its persuasiveness. For now, I assign a score of 5 but would consider raising it if the authors satisfactorily address my main concerns.

---

### Official Review · Reviewer_sS42 · 2024-11-02

**Soundness:** 3
**Presentation:** 4
**Contribution:** 3
**Rating:** 5
**Confidence:** 4

**Summary:**

This paper introduces “Decouple-then-Merge” (DeMe), a novel finetuning framework designed to enhance diffusion model training. The motivation stems from an observed issue in diffusion models: the same denoising model is used across various timesteps, leading to conflicting gradients, particularly at non-adjacent timesteps, which can degrade training quality and model performance. To address this, the authors propose decoupling the training by fine-tuning separate denoising networks over distinct timestep ranges. The authors propose three techniques during finetuning to improve performance: Consistency Loss, Probabilistic Sampling and Channel-wise Projection.  After fine-tuning, these specialized models are merged back into a single model within the parameter space, maintaining training efficiency without added inference costs. Experimental results demonstrate that DeMe enhances image generation quality across multiple benchmarks, including both unconditional (CIFAR-10, LSUN-Church, LSUN-Bedroom) and text-to-image generation tasks (Stable Diffusion).

**Strengths:**

1. The paper is well-motivated from the perspective of conflicting gradients at various timesteps. The analysis of cosine similarity and distribution of similarities across various timesteps is neat and complements the motivation in the paper.
2. Overall, the paper is very well written. The figures are extremely detailed and clearly explain the concepts/ideas in the paper.
3. The ablations and analysis in the paper is comprehensive. I particularly enjoyed the task vector (Figure 7,8) and loss landscape analysis (Figure 4, 7). The authors do a good job of breaking down the various components in the DeMe framework.

**Weaknesses:**

1. Experiments:
    1. The baseline methods that the authors compare against are Min-SNR etc. These methods are loss-reweighting strategies that are not necessarily designed for finetuning. The authors run ~80K iterations of finetuning on top of the base model (model before finetuning). Can the authors clarify if this is correct? If this is true, then I think the authors should also compare against training the entire model from scratch with Min-SNR and other baselines. Another simple baseline would be to train the base model for 80k more iterations and show the DeMe performs much better.
    2. The authors should compare against other baseline methods as well that are motivated by the similar objective.
        - Improving Training Efficiency of Diffusion Models via Multi-Stage Framework
        and Tailored Multi-Decoder Architecture [1]
        -  Not All Steps are Equal: Efficient Generation with Progressive Diffusion Models [2] — This work also operates in the training a diffusion model and then finetuning it in different sub-groups based on the timesteps.
        -  Addressing Negative Transfer in Diffusion Models [3] — This work is motivated by the same observation that training a single model for denoising at various steps is a form of multi-task learning.
    c. The results section would be stronger if the authors also report the results with more recent diffusion models such as EDM [4].
2. In L311-312, the authors mention that the optimal combination of $w_i$ is chosen via grid search. Could the authors elaborate on this? Do the authors have a validation set which they use? What is the optimal value that was used in the final results?

[1] Zhang, Huijie, et al. "Improving Training Efficiency of Diffusion Models via Multi-Stage Framework and Tailored Multi-Decoder Architecture." Proceedings of the IEEE/CVF Conference on Computer Vision and Pattern Recognition. 2024.
[2] Li, Wenhao, et al. "Not All Steps are Equal: Efficient Generation with Progressive Diffusion Models." arXiv preprint arXiv:2312.13307 (2023).
[3] Go, Hyojun, et al. "Addressing negative transfer in diffusion models." Advances in Neural Information Processing Systems 36 (2024).
[4] Karras, Tero, et al. "Elucidating the design space of diffusion-based generative models." Advances in neural information processing systems 35 (2022): 26565-26577.

**Questions:**

1. How many iterations is the base model trained for in Table 1? I could not find these details in the paper.
2. Could the authors provide intuition on why the performance of the diffusion model before merging is worse than after merging. Merging the diffusion model definitely improves efficiency by a significant margin but it’s not clear why it should improve performance. This is especially surprising to me given the dissimilarity between the task vectors.
3. From the results in Appendix F (Influence on Probability p), the experiment with P=0 suggests part of the improvement in FID on CIFAR-10 seems to be because of channel-wise projections (as well additional iterations of training, model merging). (This is in reference to P=0 performing better than base model in Figure 9). The motivation of this method was based on specific models for different timesteps. With P=0, N different models are trained separately with same set of timesteps [0, T) and then merged. It makes me wonder about the true impact of decoupling in this framework.

---

### Official Review · Reviewer_dwer · 2024-11-03

**Soundness:** 3
**Presentation:** 3
**Contribution:** 3
**Rating:** 3
**Confidence:** 4

**Summary:**

The paper presents an improved method for training diffusion models. Specifically they present it as a fine-tuning strategy for pretrained models. The strategy consists of training separate models for different time steps and then merging them into a single model that is used at inference. The authors show that the gradients of the denoising network at different timesteps can have different gradients which can result in interference during training. Their strategy leads to improved FID in different datasets.

**Strengths:**

- The method is sound and provides a better understanding of diffusion models at different timesteps
- The authors provide extensive experimentation to prove that this technique can be beneficial

**Weaknesses:**

The paper itself is very nice but it lacks completeness and the baselines are weak. I present some of these points here:
- How was figure 1 built? Did you use the same images across different time steps, or did you vary the image with the time steps? Also how many samples were used in the construction of this graph
- In the channel-wise projection section, what exactly do you mean by channel-wise mapping vs spatial-mapping? This is never explained and it is not clear what it should be
- During the merging stage you use grid search to find the values of $w_i$, what values of $w_i$ do you consider? It seems like the grid search would grow exponential in $N$ so it becomes important to know how did you finetune this parameter? Did you run FID on all the possible values of w_i, if so, how many samples did you use?
- Could you please explain figure 4, the neural network is very high dimensional but we are somehow visualizing the loss landscape in only 2 dimensions, how is this possible?
- I feel like we are considering very weak baselines. For instance the seminal work of EDM [1] showed a loss reweighing scheme that leads to quite impressive results, this was further investigated in [2] were an extensive research was done to find the optimal. Even on the theoretical front [3] has shown that the choice of EDM is optimal. It seems very important to compare to these works as any modern diffusion model is based on these techniques. It would also be interesting to see if there models can be further improved through this technique


[1] Karras, Tero, et al. "Elucidating the design space of diffusion-based generative models." Advances in neural information processing systems 35 (2022): 26565-26577.
[2] Kingma, Diederik, and Ruiqi Gao. "Understanding diffusion objectives as the elbo with simple data augmentation." Advances in Neural Information Processing Systems 36 (2024).
[3] Wang, Yuqing, Ye He, and Molei Tao. "Evaluating the design space of diffusion-based generative models." arXiv preprint arXiv:2406.12839 (2024).

**Questions:**

See weaknesses, it would be great if the authors can address these

---

### Official Review · Reviewer_zsSP · 2024-11-05

**Soundness:** 2
**Presentation:** 2
**Contribution:** 2
**Rating:** 3
**Confidence:** 4

**Summary:**

This paper presents a novel framework called Decouple-then-Merge (DeMe) to address the issue of gradient conflict during diffusion model training across different timesteps. DeMe addresses such an issue by finetuning separate models for different timesteps to minimize negative interference, then merging them back into a single model to retain efficiency during inference. Three key techniques are introduced during the fine-tuning stage: consistency Loss, probabilistic sampling, and channel-wise projection, all aimed at preserving knowledge sharing across timesteps while allowing for specialization. After fine-tuning, the separate models are merged into a single model in the parameter space, combining the benefits of specialized training without incurring additional computational or storage costs during inference. Experiments across various datasets show that DeMe significantly improves image quality for text-to-image generation tasks, achieving better performance than the baseline loss reweighting methods.

**Strengths:**

- The paper tackles a timely and practically-relevant problem supported by a fair amount of experiments. Negative transfer across timesteps in diffusion models is an area with limited prior research, making this work particularly valuable.
- Overall, the paper is clearly written and easy to follow.

**Weaknesses:**

- A primary weakness of this paper is the lack of experimental setups, particularly regarding scalability to large-scale and high-resolution data. To better evaluate DeMe’s effectiveness across different model architectures and training strategies—such as score matching versus flow matching loss—additional experiments on recent diffusion models, including MDT, SDXL, and EDM, would be valuable.
- The negative transfer phenomenon is a central topic in multi-task learning (MTL) literature, and recent works [1, 2] have analyzed diffusion models from this perspective. Why did the authors use only three loss reweighting baselines, instead of incorporating MTL methods like those in [1]?
- Given the complexity of DeMe’s components, more ablation studies and an analysis of its robustness to hyperparameter settings would be beneficial. For instance, could the authors provide an extended ablation study using larger datasets and models beyond what is shown in Table 3? Additionally, regarding Appendix F, how should DeMe determine the number of intervals $N$? It also appears that the training interval sampling probability $p$ may not be a critical contributing factor, as shown in Figure 9. Please correct me if I am wrong.
- Regarding the loss landscape analysis, for example in Figure 7, please provide further experimental details as to which training loss is used and the definition of two tasks being visualized. In Figure 4, the leftmost figure shows a pre-trained diffusion model, trained on the full timestep range, exhibiting high loss (in red) on the CIFAR-10 training loss landscape. Why is this the case? Furthermore, even if the pre-trained diffusion model is situated at a high-loss critical point, it is unclear why merging multiple task-specific fine-tuned models tends to lead the model to a lower-loss region. It would be helpful if the authors can provide further theoretical insights.

[1] Go et al., “Addressing negative transfer in diffusion models”, NeurIPS 2023.\
[2] Moon et al, “A Simple Early Exiting Framework for Accelerated Sampling in Diffusion Models”, ICML 2024.

**Questions:**

- Regarding channel-wise projection, to my understanding, DeMe employs additional trainable projection matrices. How does this affect total cost in terms of inference speed, memory cost, etc?
- There is a watermark “shutterstock” at the leftmost figure in Figure 5.

---

### Author Response · Authors · 2024-11-15
**Thank you to all the reviewers.**

Dear reviewers,


We appreciate all of you for taking the time to review our work. We decided to withdraw the paper for revision. We will revise our manuscript based on your reviews, adding baselines and related methods for much more abundant experiments. We are also planning to release our code and pre-print paper. Thank you very much for your review and welcome to keep following our work!


Best Regards,

Submission1755 Authors.

---

### Note · Authors · 2024-11-15

I have read and agree with the venue's withdrawal policy on behalf of myself and my co-authors.